# Pathways and Estimate of Aquifer Recharge in a Flood Basalt Terrain; A Review from the South Fork Palouse River Basin (Columbia River Plateau, USA)



**Giacomo Medici [1,*] and Jeff B. Langman [2]**

[1] Department of Earth Sciences, Sapienza University of Rome, Piazzale Aldo Moro, 5, 00185 Rome, Italy

[2] Department of Earth and Spatial Sciences, University of Idaho, Moscow, ID 83844, USA

\* Correspondence: g.medici.gr@gmail.com

**Abstract:** Aquifer recharge is one of the most important hydrologic parameters for understanding available groundwater volumes and making sustainable the use of natural water by minimizing groundwater mining. In this framework, we reviewed and evaluated the efficacy of multiple methods to determine recharge in a flood basalt terrain that is restrictive to infiltration and percolation. In the South Fork of the Columbia River Plateau, recent research involving hydrologic tracers and groundwater modeling has revealed a snowmelt-dominated system. Here, recharge is occurring along the intersection of mountain-front alluvial systems and the extensive Miocene flood basalt layers that form a fractured basalt and interbedded sediment aquifer system. The most recent groundwater flow model of the basin was based on a large physio-chemical dataset acquired in laterally and vertically distinctive locations that refined the understanding of the intersection of the margin alluvium and the spatially variable basalt flows that filled the basin. Modelled effective recharge of 25 and 105 mm/year appears appropriate for the basin's plain and the mountain front, respectively. These values refine previous efforts on quantifying aquifer recharge based on Darcy's law, one-dimensional infiltration, zero-flux plane, chloride, storage, and mass-balance methods. Overall, the combination of isotopic hydrochemical data acquired in three dimensions and flow modelling efforts were needed to simultaneously determine groundwater dynamics, recharge pathways, and appropriate model parameter values in a primarily basalt terrain. This holistic approach to understanding recharge has assisted in conceptualizing the aquifer for resource managers that have struggled to understand aquifer dynamics and sustainable withdrawals.

**Keywords:** aquifer recharge; sustainability water resources; groundwater isotopes; groundwater flow modelling

## 1. Introduction

Groundwater recharge represents the replenishment of aquifers, and it is one of the most important hydrologic parameters for managing and protecting water resources from over-exploitation and contaminant intrusion [1–5]. Groundwater recharge represents the connection between the land surface and the aquifer and reflects the climate, vegetation, land use, and vadose zone characteristics of a given area [6–8]. Despite its importance, groundwater recharge is seldom addressed when characterizing aquifer systems and often is simply used as an adjustment parameter for groundwater model calibration.

The habit of model fitting recharge by manual calibration or determination by Model-Independent Parameter Estimation (PEST) is a paradox since groundwater flow models are primarily sensitive to recharge rather than hydraulic conductivity [9,10]. Experimental efforts to constrain natural recharge should be equal or superior to constraining hydraulic conductivity.

Hydraulic conductivity can be determined by aquifer tests that can provide the necessary information to calibrate groundwater flow models [11–16]; while aquifer recharge

commonly is adjusted to fit potentiometric surface fluctuations. The quantification of recharge is challenging because of the substantial lack of measurements that constrain seasonal or annual volumes of water moving from the land surface to the saturated zone. Therefore, groundwater recharge typically is indirectly estimated using a variety of methods, formulas, and parameters at different spatial and temporal scales that have substantial uncertainties [17,18]. Yet, recharge rates can be constrained, and reliable recharge volumes can be achieved through an intersection of hydraulic and geochemical perspectives. The use of hydrologic tracers (e.g., isotopes, noble gases) may allow for determination of applicable recharge rates across different terrains and land use types that can be tested within a flow model's domain [19].

Our review of historical and recent attempts to unravel flowpaths/traveltime/recharge in the basalt terrain of the South Fork Palouse River Basin (Figure 1) is meant as a holistic interpretation of system dynamics for understanding recharge in contrast to common aquifer characterization that often focuses on a primary discipline (e.g., groundwater levels and modelling vs. hydrologic tracers) (e.g., [20–26]). The review of past and current attempts at understanding recharge in the South Fork Palouse River Basin is to highlight the intersection of hydraulic and hydrogeochemical methods that allowed for a greater understanding of aquifer dynamics in a basin where scientists have struggled to provide implementable ideas to resource managers for sustainable groundwater use.

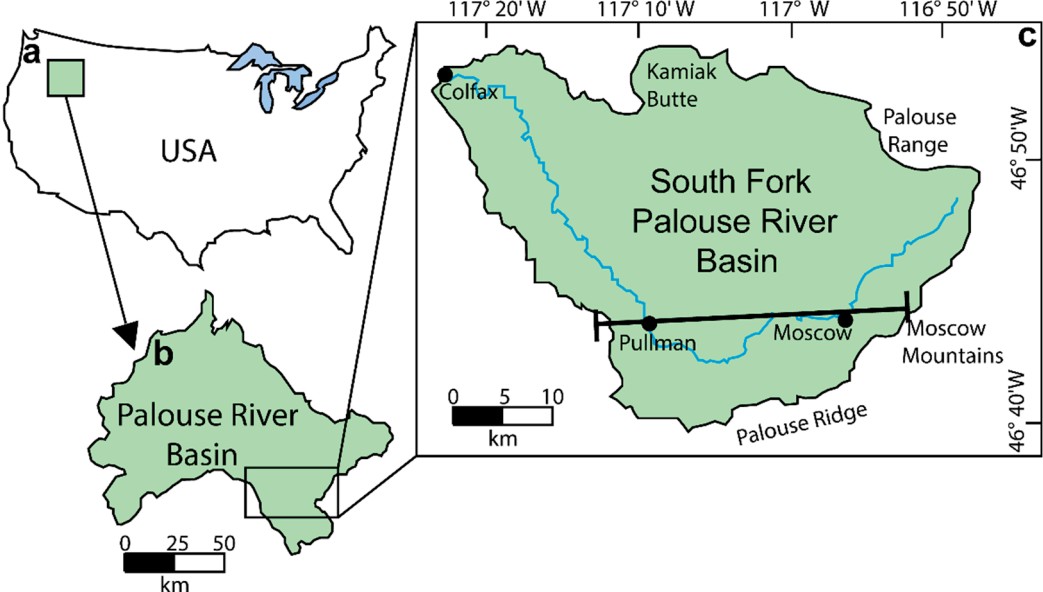

**Figure 1.** Study area. (**a**) Location of the study site in the United States of America, (**b**) South and North Fork Palouse River basins (35), and (**c**) South Fork Palouse River Basin.

In this research, we review the last 50 + years of literature on both quantifying aquifer recharge and constraining the recharge source in this complex basin that is composed of fractured-basalt flows and interbedded sediments [27]. The South Fork Palouse River Basin of the Columbia Plateau (Figure 1) is characterized by a Mediterranean climate (Köppen climate classification of Mediterranean-influenced warm-summer humid continental climate) and low infiltration/percolation rates through the basalt flows that compose the majority of the subsurface stratigraphy [28]. The basin's groundwater resources have been subjected to overexploitation with declining groundwater levels since the 1930s [29]. As a consequence of this continuing groundwater mining, substantial investigation of the aquifer has occurred by water resource managers, government scientists, and university researchers, which has made the basin an excellent laboratory to evaluate the intersection of hydraulic and hydrogeochemical interpretations of aquifer dynamics.

Notably, a substantial portion of research efforts in this basin has focused on soil and surface water (e.g., [30–34]), or the vadose zone (e.g., [35–38]). Yet, the loess soils and vadose zone sediments are substantially different than the >1000 m of fractured basalt and interbedded sediments composing the aquifer system (Figure 2) [28]. This traditional view of examining surface water dynamics to estimate aquifer recharge can provide the conceptual framework for infiltrating water, but in basins where subsurface stratigraphy differs so greatly from surface geology, the subsurface characterization must be a priority. Within the basin, a recent focus on subsurface geology motivated a re-examination of potential recharge processes and the need for additional hydrologic tracers to be incorporated into any new groundwater flow model [27,28].

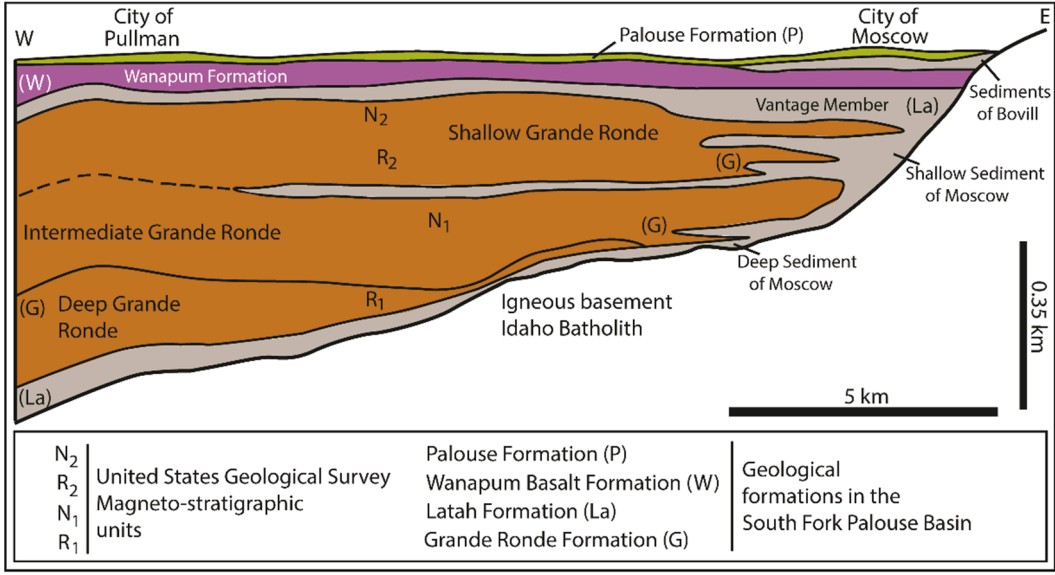

**Figure 2.** Schematic geological cross-section of the Columbia River Plateau Regional Aquifer System in the South Fork Palouse River Basin (modified from [27,28]).

In summary, combining surface hydrologic data, a robust three-dimensional hydrogeologic characterization of the physico-chemical aquifer properties, and modelling efforts, this review aims to illustrate the advantage of a holistic approach to the understanding of aquifer recharge in heterogeneous systems such as the fractured basalt and interbedded sediments of the Columbia River Plateau. Specific research objectives are (i) describe the methods that assisted in defining the principal recharge source of the basin, (ii) compare different recharge rates derived from seven different physico-chemical methods, (iii) identify the most successful approach to quantify recharge rates to inform groundwater flow and particle tracking models, and (iv) address future research needs in this heterogeneous system.

## 2. Materials and Methods

### 2.1. Geology

The South Fork Palouse River Basin is located on the eastern margin of the Columbia River Plateau, covering an area of about 290 km$^2$ that is largely dedicated to farming by cultivating plants and livestock (Figure 1a–c). The study area is characterized by a Cretaceous-Paleogene igneous basement (Idaho Batholith) overlain unconformably by the Miocene flood basalts of the Columbia River Basalt Group and interbedded sediments of the Latah Formation (Figure 2). The flood basalt is formally divided into the Wanapum and the Grande Ronde formations in the basin (Figure 2; Table 1). The interbedded sediments of fluvio-lacustrine origin are part of the Latah Formation (Figure 2), which is organized into four members (see Table 1 for geological nomenclature). The Miocene flood basalts are interbedded with alluvium deposited between basalts flows (Figure 2). The fractured

basalts and interbedded sediments are unconformably overlain by Quaternary aeolian deposits of the Palouse Formation (loess) (Table 1; Figure 3a–c). The Miocene flood basalts entered the South Fork Palouse River Basin from the southwest, filling the pre-existing paleo-topography though volcanic eruptions from multiple vents [27,28]. The periods of volcanic quiescence between basalt flows allowed for deposition of alluvial deposits that were buried and preserved as interbeds throughout the volcanic sequence of basalts [39–42]. These interbeds of sedimentary origin show a maximum thickness of ~35 m to the east and can pinch out to the west (Figure 2). The variation of the spatial extent of the various basalt layers and discontinuity of the interbedded sediments creates a highly heterogeneous aquifer with large variations in vertical and horizontal permeability. The sedimentary interbeds are dominated by clay and silt deposited in the floodplain, although sandy fluvial channels have been identified [43], which can provide primary pathways for recharge and groundwater [44]. From a volcanological point of view, this part of the Columbia River Plateau is characterized by systems of dikes which are oriented NW–SE to NNW–SSE that were the feeders of the basalt floods [28,41].

**Table 1.** Hydro-stratigraphy at the study site based on interpretation of core logs [27,28] with detail of the lithologies for the deposits of the Palouse Formation (P) and Latah (LF) Formation and the Columbia River Basalt Group (CRBG).

| Lithostratigraphy | Geological Time | Thickness (m) | Lithotype |
|---|---|---|---|
| Palouse Formation, P | Holocene-Pleistocene | 1–10 | Clay, Silt |
| Sediments of Bovill Member, LF | Miocene | 0–20 | Clay, Silt, Sand |
| Wanapum Basalt Formation, CRGB | Miocene | 40–70 | Fractured Basalt |
| Vantage Member, LF | Miocene | 2–10 | Clay, silt, sand |
| Upper Grande Ronde Formation, CRGB | Miocene | 50–80 | Fractured Basalt |
| Shallow Sediments of Moscow Member, LF | Miocene | 0–15 | Clay, silt, sand |
| Intermediate Grande Ronde Formation, CRGB | Miocene | 160–220 | Fractured Basalt |
| Lower Grande Ronde Formation, CRGB | Miocene | 20–80 | Fractured Basalt |
| Deep Sediments of Moscow Member, LF | Miocene | 0–25 | Clay, silt, sand |

In addition to the layers of basalt and interbedded sediments of the basin, a gentle anticline that may reflect the underlying basement and subsequent tectonic forces is oriented NNW-SSE and plunges towards the NNW across the southern part of the basin. Emplacement and cooling of the basalt along with subsequent tectonic forces induced high-angle (70°–90°) joints that are present in all of the different basalt flows. Basalt bedding plane discontinuities are sub-horizontal (Figure 4), mainly dipping towards the northwest, and have shown a maximum 5° dip in quarries that were studied by [43].

### 2.2. Surface Hydrology and Hydrogeologic Units

The South Fork Palouse River Basin is located along the Palouse Range in the larger Palouse River Basin that is part of the regional Columbia River Basin (Figure 3). The South Fork Palouse River Basin has an eastern margin bounded by the Palouse Range, Tomer Butte, and Paradise Ridge along with low-lying hills that resulted from intrusion of the Idaho Batholith [43]. The central and western portions of the basin are dominated by rolling hills of the Palouse; a region created by the uppermost basalt flow and subsequent deposition of alluvial and aeolian material (Figure 3a,b). The region's climate is driven by its proximity to the Pacific Ocean and the northern Rocky Mountains, which produces a winter maritime climate and a summer continental climate [34,45]. Annually, the mean temperature is 9.1 °C. The combination of a summer Mediterranean weather, the extensive rolling hills, and intense framing activity has led the Palouse to be termed the Tuscany of America (Figure 3a).

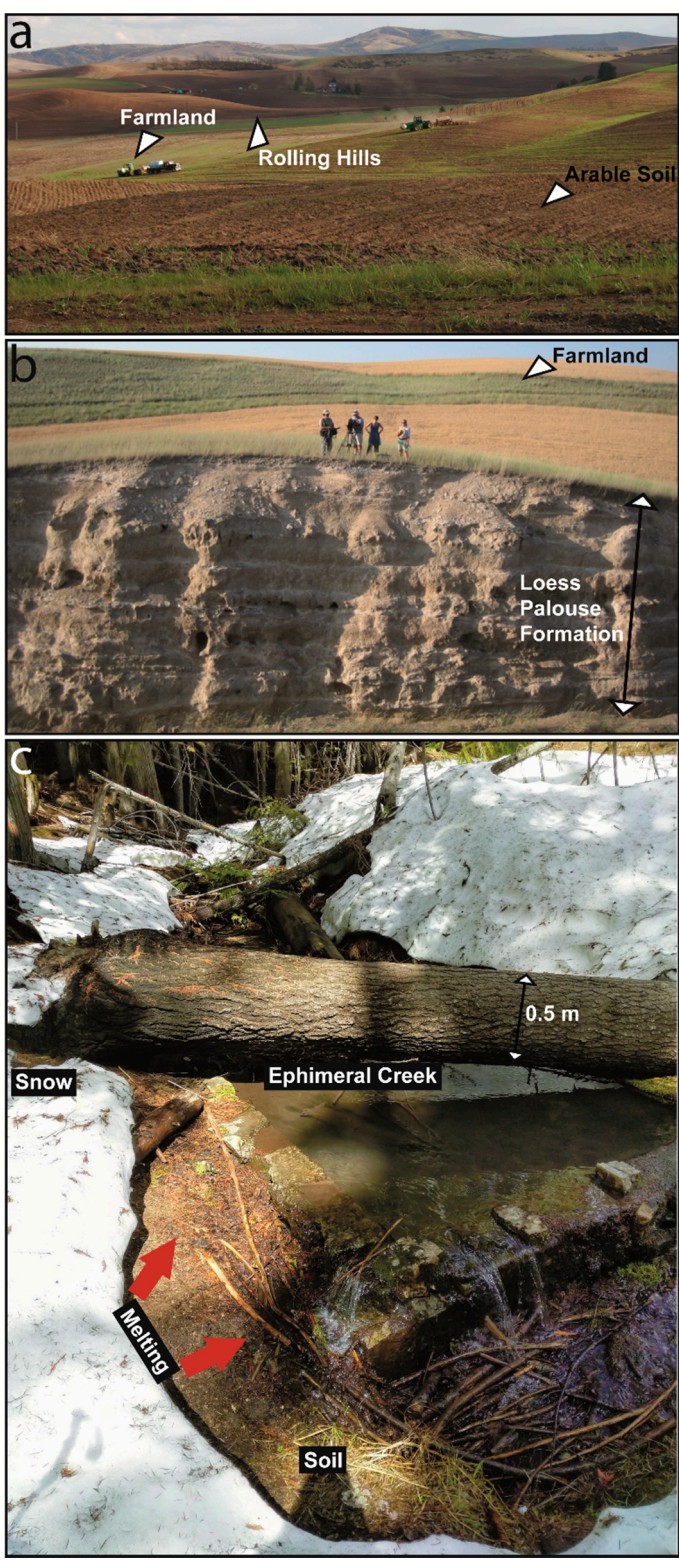

**Figure 3.** Landscape and loess common to the South Fork Palouse River Basin. (**a**) Landscape with rolling hills, (**b**) Loess cover (from symphonyofthesoil.com), (**c**) Soil and springtime snowmelt filling a cistern built into an ephemeral creek in the headwaters of the Palouse Range.

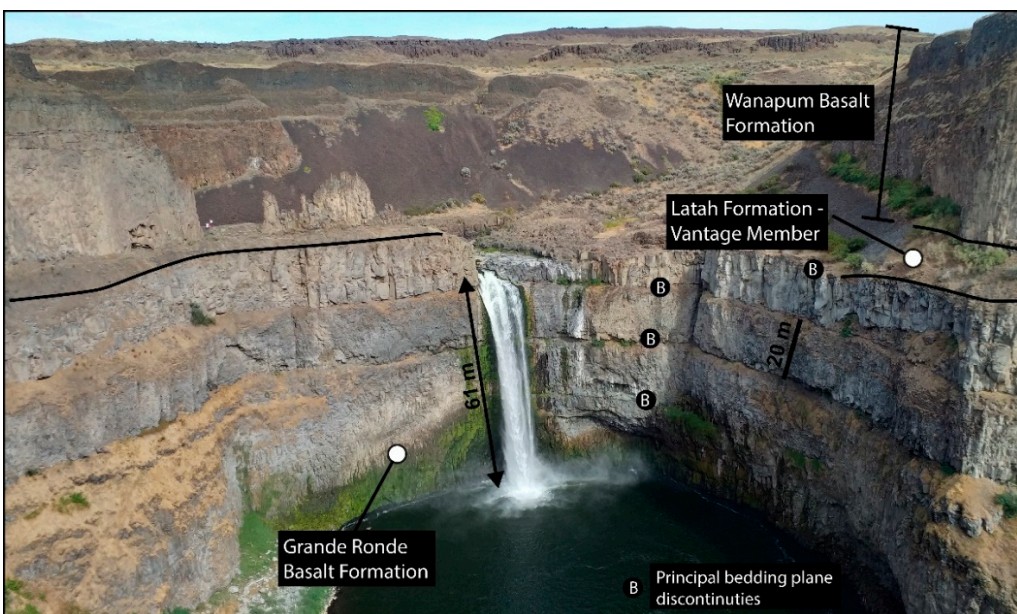

**Figure 4.** Outcrop exposure of the Latah, Wanapum, and Grande Ronde formations of the Columbia River Plateau Regional Aquifer System at the Palouse Falls with bedding plane fractures (B) in evidence.

The South Fork Palouse Basin receives approximately 60 cm of precipitation (water equivalent), including 126 cm of snowfall. The Palouse Range snowpack averages a snow-water equivalent of 50 cm in thickness at its highest elevations. The snowpack will develop in late fall and typically last until late spring. Ephemeral creeks will respond to snowmelt and precipitation in winter/spring and may contain flow into summer (Figure 3c). These systems, along with subsurface water, feed downgradient intermittent and perennial creeks along with providing recharge to the aquifer system of the basin. The hydrologic recharge source that is the Palouse Range snowpack has produced variable isotope signals in connected surface-water and groundwater systems [45,46].

The South Fork Palouse River Basin aquifer system is composed of unconfined and primarily confined saturation zones found in the upper sediments, basalt flows, and interbedded sediments with an unconfined alluvial aquifer along the eastern basin margin. The Plio-Quaternary cover (Figure 3a,b) consisting of the loess (Palouse Formation) and sediments of Bovill (Latah Formation) have been hydraulically characterized (K = 0.001–2 m/day; n = 111) through a relatively large number (n) of slug (n = 55) and Guelph permeameter (n = 56) tests [47–51]. The basin creek/river system is relatively stable and has not completely eroded through the relatively thick (10–90 m thick according to water-well and core logs), low-permeability cover in most locations across the basin plain. This low-permeability cover contains argillitic soil and clay layers that can act as restriction to vertical flow and may produce translatory flow that is directed towards the streams [35]. Recent investigations have suggested high conductivity flowpaths from the mountain front of the Palouse Range through the unconfined alluvial aquifer into the confined aquifer system of the fractured basalts and interbedded sediments [44,51]. These paleochannels are characterized by higher porosity and moisture content that can be traced from the mountain front to the city of Moscow and are stratigraphically assigned to the sediments of Bovill, Vantage Member, and sediments of Moscow of the Latah Formation [28,35]. Notably, analysis of pumping tests from deeper wells (100–600 mBGL) indicated transmissivity one order of magnitude higher in Moscow compared to Pullman [52–54]. This lateral variation in transmissivity reflects the transition of the thicker alluvium and interbedded sediments near the mountain front and their eastward thinning [43].

The thicker mountain-front sedimentary sequence and associated paleochannels represent preferential flow pathways for snowmelt from the Palouse Range and likely are primary routes into the deeper portions of the confined aquifer system [44,45]. The princi-

pal unit of the confined aquifer system that has been exploited for municipal water is the Grande Ronde Formation, which is characterized by hydraulic conductivity ranging from 6 up to 22 m/day [52–54]. This aquifer unit is the larger of the two primary formations (other being the upper Wanapum Formation) and contains transmissive zones associated with the interbedded sediments (layer horizontal flow during aquifer testing) that connect this relatively deep portion of the aquifer to the alluvium of the mountain front. Note that, horizontal hydraulic conductivity ($K_h$) is defined as the ratio of the transmissivity and screen lengths, i.e., high ($K_h/K_v \sim 10^2$–$10^3$) flow anisotropy of the fluvio-volcanic deposits of the Columbia Plateau Regional Aquifer System (CPRAS) [55,56]. The Grande Ronde Formation is ~450 m thick, resides about ~200 m below the basin surface, and is divided into lower, intermediate and upper units based on the occurrence of interbedded sedimentary layers and magneto-stratigraphic studies (Figure 2). Hydraulic conductivity of the CPRAS typically declines sharply below ~600 m below the ground surface in the [57]. This permeability threshold approximately occurs at the study site at the top of the lower Grande Ronde Formation (Figure 2). However, the dominance of $K_h$ in the basin shifted the focus of recharge of the confined aquifer system to a horizontal perspective instead of the traditional vertical pathways.

A prior view of primarily vertical recharge in the South Fork Palouse River Basin was sustained into the 2010s because of $^{14}$C isotopes that indicated significant vertical age variation and potential Pleistocene water in the South Fork Palouse River Basin [58,59]. Yet, previous groundwater flow models included vertical flow anisotropies ($K_h/K_v$) ranging from 2000 up to 5500 in the larger CPRAS [55,56]. Previous flow models were realized at the study site using an Equivalent Porous Medium (EPM) approach that confirm a high-flow anisotropy ($K_h/K_v$ = 90–7000) fitting the lateral continuity of the bedding planes at the Palouse Falls as shown in Figure 4 [60,61]. Such numerical models indicate groundwater flow directed towards the west and distinguish shallow (<150 mBGL) and deep (150–900 mBGL) aquifer systems from the mountain front to Colfax (Figure 1).

## 3. Aquifer Recharge

### 3.1. Pathways

Past research has suggested very limited recharge and paleowaters in the aquifer system [59], spatially uniform recharge across the basin floor [61–63], focused recharge around the creek system [31,64–66], and a substantial mountain-front recharge along the basin's eastern margin [45,51]. Hydrologic tracer analysis of groundwater collected at different localities and depths revealed the primary recharge source of snowmelt from the eastern margin in the South Fork Palouse River Basin (Figures 1 and 3c), which aligned with updated geologic perspectives of the relation of mountain-front alluvium and the fractured basalts and interbedded sediments of the confined aquifer system [43]. Previous research discriminated source waters, recharge pathways, and alteration of the redox environment along with identifying contamination of prior $^{14}$C analysis from mantle $CO_2$. A shallow (<150 mBGL) portion of the aquifer system was characterized as containing a mix of mid-basin surface water, mountain-front runoff, and snowmelt as indicated by different source-water signals identified by $\delta^{16}$O, $\delta^{18}$O, $\delta^{13}$C, $\delta^2$H, and noble gas values [51]. This mixed distribution of source waters in the upper aquifer arises from the mixing of rapidly recharged snowmelt and more slowly recharged surface water. This source water mixing is further indicated by the wide range of groundwater temperatures (12–19 °C), and estimated recharge temperatures determined from noble gas analysis [67]. In contrast, at greater depths (>150 mBGL) in the basin, $\delta^{18}$O, $\delta^{13}$C, and $\delta^2$H values are in a narrow range that align with the isotope signal of the mountain-front snowmelt [51]. Of note, Ref. [67] attempted to re-interpret prior $^{14}$C ages because of mantle gas inputs and found groundwater ages of ~$10^6$ days (or ~$10^4$ years) in the deeper aquifer that were 64% smaller than previous estimate [59].

The most recent research on the deep wells of the aquifer system shows how $\delta^{18}$O and $\delta^2$H values of the groundwater overlap those of the Palouse Range snowmelt and

its pathways through the mountain-front alluvium to the confined aquifer system [45,46]. This recent research confirms a conceptual framework of snowmelt infiltration and runoff as principal recharge source by comparing the isotope fingerprint of the Palouse Range snowpack, snowmelt, ephemeral and perennial creeks, and groundwater at different depths. Such a change in recharge perspective began to align with the known high $K_h/K_v$ values used in prior flow models and shifted the focus from uniform, distributed recharge to horizontal recharge pathways that corresponded to the increased understanding of the geologic setting.

### 3.2. Estimation

Over the past 50 years, the confined aquifer system of the South Fork Palouse River Basin has been the object of considerable research efforts, which produced a large range of estimates of potential (0–317 mm/year) and effective (0.4–103 mm/year) recharge (summarized in Table 2). Potential recharge typically is higher than effective recharge since we generally do not fully incorporate water retention in the vadose zone during recharge estimates, being strongly or strictly dependent from precipitation and evapotranspiration [6]. Two soil moisture routing (SMR) models of potential recharge were proposed in the study area by [31,35] (Table 2). Both models were grid-based and simulated interception, evapotranspiration, subsurface lateral flow, percolation, and surface run-off. The model [31] was an update to the SMR model [35] that covered a larger spatial scale. Output of the SMR models indicated relatively high values of potential recharge (201–317 mm/year) given the focus on the potential recharge from snowmelt/surface runoff from the higher elevation areas of the eastern basin that accumulate substantial snowpacks. Additionally, the potential origin of highly permeable paleochannels along the mountain-front correspond to the large source of potential recharge from the mountain snowpack. This focus on surface pathways along the mountain front and potential connection to groundwater recharge began to shift the prior view of spatially uniform infiltration/percolation towards the potential of dominant recharge along the eastern margin of the basin.

**Table 2.** Recharge values vs. methodology for the Columbia River Plateau Regional Aquifer System in the South Fork Palouse River Basin.

| Type of Recharge | Method | Recharge Rates (mm/Year) | Reference |
|---|---|---|---|
| Potential | Soil Moisture Model | 0–317 | [35] |
| | LEACHIM | 105 | [72] |
| | LEACHIM | 25–103 | [73] |
| Effective | Mass Balance | 16 | [68] |
| | Mass Balance | 30 | [69] |
| | Zero Flux Plane | 45 | [74] |
| | Chloride | 3–40 | [70] |
| | Chloride | 3–10 | [38] |
| | 3D MODFLOW Model | 250 | [60] |
| | 3D MODFLOW Model | 25–103 | [61] |
| | 3D Fortran Model | 17 | [61] |
| | Storage Equation | 122 | [71] |

The effective recharge (value fully incorporating water retention processes in the entire vadose zone) of the aquifer system in the South Fork Palouse River Basin was previously determined by water balance [68,69], chloride mass-balance [38,70], storage equation [71], one-dimensional infiltration [72,73], and zero flux plane [74]. One-dimensional infiltration-models using LEACHM (the Leaching Estimation and Chemistry Model) produced recharge values ranging from 25 to 103 mm/year. The water balance method was used to constrain effective recharge and produced values ranging from 16 to 30 mm/year

(Table 2). However, the annual recharge rate was estimated at 122 mm, and derived from the Equation (1), which assumed that the storage is equal to the change of water level:

$$R_v = S \times A \times \Delta h + P. \tag{1}$$

where $R_v$ is the recharge volume, S is the storativity known from pumping tests analyzed by [52], A is the area of the South Fork Palouse Basin, $\Delta h$ variation in piezometric surface, and P is the volume of water pumped. Such attempts to integrate a storage component produced a wide variation in recharge calculations because of the difficulty in applying S coefficients derived from individual wells across this highly heterogeneous aquifer system. An aquifer recharge rate also was estimated at 45 mm/year using the zero-flux plane method and field tensiometers. Application of the chloride mass-balance method indicated potential recharge rates ranging from 2 to 30 mm/year (Table 2). In the latter method, the effective recharge was extrapolated by comparing Cl- concentration in rainfall and groundwater samples collected from wells in across the basin [18].

*3.3. Groundwater Flow Models*

Multiple groundwater flow models were developed for the South Fork Palouse River Basin. The earliest mathematical model was created by [75], who applied the [76] formulation of storage to the basin. This early calculation of groundwater flow assumed no recharge to the basin based on the conclusion that only 10% of annual pumping was recent groundwater replenishment [77]. In contrast, Ref. [61] created a FORTRAN numerical flow model of the Grande Ronde aquifer to represent Darcian flow in a 3D finite element grid. This computer model identified vertical leakage as the primary recharge pathway that was constrained by a recharge rate of 17 mm/year (Table 2).

New modelling efforts [78] led to update Barker's model to include the Wanapum Formation and loess in the aquifer system while adding the deep percolation model developed by the U.S. Geological Survey to estimate continuous vertical recharge (spatially uniform infiltration) of 63 mm/year.

Following [78] attempt to include vertical recharge to the entire basin, Ref. [60] built a MODFLOW model and refined the thickness of the Grande Ronde basalt layers but kept the same three primary units of Palouse Formation (loess), Wanapum basalt, and Grande Ronde basalt. Ref. [79] refined the [60] model for the Moscow-Pullman area with no attempt to refine the recharge estimate (63 mm/day) and indicated large uncertainty in recharge and flow into the Grand Ronde layer. Recommendations on conservation and municipal pumping to stabilize the aquifer system were provided to the Palouse Basin Aquifer Committee (consortium of regional water operators). These suggestions were based on the Lum model [60], but the model was deemed to have a substantial degree of uncertainty due to paucity of a robust dataset at that time. As a consequence of the lack of a valid modelling tool, groundwater levels continued to decline over the years. Given the continued uncertainty regarding recharge to the aquifer system in the basin; an attempt was made in 2005 to physically identify recharge pathways by drilling 53 boreholes (7 fully and 45 partially penetrating the sedimentary cover) through the loess and sediments of Bovill. Yet, the investigators concluded very limited recharge may occur in the eastern portion of the basin given the low permeability of the sediments [80,81].

Given the perceived failure to understand recharge and aquifer dynamics in the South Fork Palouse River Basin, researchers attempted to apply hydrologic tracers to understand source waters and recharge pathways into the confined aquifer system from 2010 to 2019. This research overlapped with efforts to better understand the geologic setting of the basin. Conceptual models that assumed Pleistocene water and limited pathways into the deeper portions of the aquifer were abandoned in favor of models that valued a primarily eastern margin snowmelt-dominated recharge zone [59]. These updated attempts to understand recharge in the basin shifted the perception of diffuse, spatially uniform infiltration and vertical recharge to horizontal recharge from the alluvial-mountain front zone. This shift in

the perception of recharge aligned with recent investigations focused on soil moisture and flowpaths between the creek system and upper sediments (e.g., [16,31,35]).

The recent efforts to understand recharge from the alluvial-mountain-front zone spurred an attempt to fit such a conceptual model into a numerical flow model for predicting aquifer dynamics. In this framework, Ref. [82] developed a new MODFLOW model that discriminated between western and eastern basin recharge. Focus was placed on the eastern snowmelt-mountain front recharge (105 mm/year) when compared to a more limited western vertical recharge (25 mm/year) that accounted for potential infiltration along the creek/river system. This model incorporates the revised conceptual view of a snowmelt-driven aquifer system derived from five years of spatially diverse hydrologic tracer data collected from (i) surface source waters (e.g., snowpack, snowmelt, creeks), (ii) shallow groundwater in the mountain-front alluvium, and (iii) deeper groundwater in the fractured basalt and interbedded sediments of the basin. Additionally, Ref. [82] incorporated the results of [38,50], who used chloride to estimate recharge rates into the aquifer system.

This shift in perception of recharge enables more reliable groundwater management recommendations than previous models, such as [60], which focused on vertical recharge. The aerially distributed recharge rate (63 mm/year) used by [60] was relatively small for the eastern margin of the basin, but its application across the entire basin produced a relatively high recharge volume; Indeed, 30% more water is applied as recharge flux compared to the [82] conceptual model of primarily eastern margin recharge. This overestimation resulted from a perception of uniform vertical recharge that did not discriminate among the limited vertical pathways across the basin.

As a consequence of the better understanding of the aquifer geology and results of the hydrologic tracer studies, the new modelling effort by [82] also aligned with prior work on effective recharge from chloride, zero flux plane, and mass-balance methods (see Table 2). Additionally, backwards unreactive particle tracking was modelled with MOD-PATH (12% effective porosity) to test the horizontal recharge conceptual model for the confined aquifer system. Average travel times ($\sim10^5$–$10^6$ days) computed by numerical modelling at the basin scale overlapped with groundwater ages ($\sim10^6$ days) calculated on samples collected on discrete points by [67]. Such travel times fit the portion of the new recharge conceptual model's focus on highly porous paleochannels as principal flow pathways as depicted in Figure 5. This approach shifts the emphasis from the mountain front into the interbedded sediments as opposed to fractured igneous rocks (e.g., basalts) that are typically characterized by much lower values ($\sim0.01$–0.1%) of effective porosities [6,83].

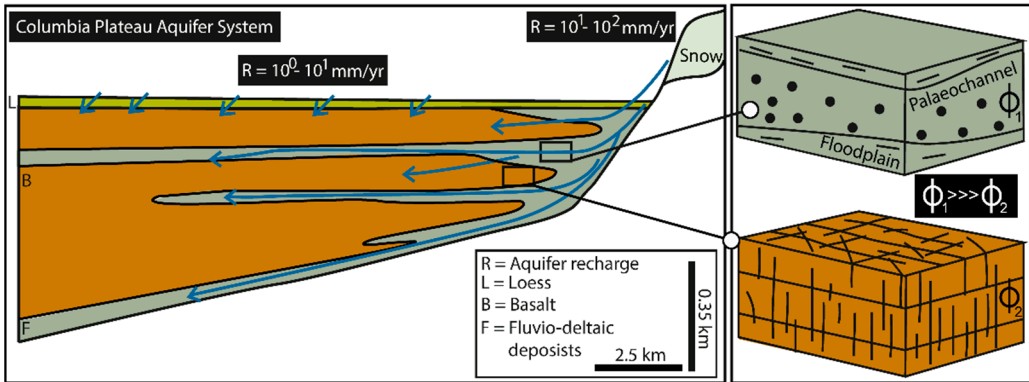

**Figure 5.** 3D hydrogeological conceptual scheme of the units of the Columbia River Plateau Regional Aquifer System in the South Fork Palouse River Basin.

The new recharge conceptual model and new groundwater flow model application contrasts with to the prior perception of diffuse vertical recharge through the basaltic

matrix that is highly un-conductive (Kcore-plugs < 0.1 m/day). In this hydraulic scenario, dominant diffusive flow is unlikely even at large (~$10^6$ days) time scales [84].

*3.4. Future Research*

The most recent version of the model is based on a large physico-chemical dataset acquired in laterally and vertically distinctive locations in the basin [82]. Therefore, values of effective recharge of 25 and 105 mm/year seem to be appropriate for the plain and the mountain front, respectively (Figure 5). The latter flow and particle tracking model can be refined by increasing recharge rates in correspondence of the Palouse River in the plain.

However, this implementation is recommended, but not crucial since the Palouse River does not completely erode the impermeable substratum of the Palouse Formation (loess in Figure 3b) and Sediments of Bovill Member in the studied basin. The river that does not play a key role on recharging the aquifer and also presents a distinctive isotope fingerprint with respect to the shallowest aquifer system in the study area [43,59]. MIKE-SHE is a numerical solution built on coupled physics-based models for overland flow, unsaturated flow, groundwater flow, and fully dynamic channel flow, that can be used in an EPM framework (same approach used by [82]) to refine dynamics of aquifer recharge at the study site by [85]. Indeed, much more information has been acquired on overland flow and the unsaturated flow with conceptual links on aquifer recharge in Palouse that need be numerically tested at the watershed scale (e.g., [30,86]).

The above-mentioned numerical solutions need to be supported by a more robust experimental program on the chloride method and isotopic environmental tracers on both the plain, and the topographic highs, respectively to refine recharge pathways and quantification. A higher-density groundwater monitoring network can also provide useful datasets. Notably, the latter proposal is particularly needed in the north-western sector of the basin near Colfax (Figure 1c). The flow pathways and the dynamics that enable the recharged water to leave the basin need to be understood at the study site.

## 4. Conclusions

Groundwater recharge is the most important hydrological parameter for understanding aquifer dynamics and the ability to sustainably manage groundwater resources. Estimating recharge is difficult because of its diffuse nature and the large task of trying to directly measure recharge pathways and estimating recharge through storage changes and models can have substantial uncertainties. Our review of the history of recharge investigations and groundwater flow model development in the South Fork Palouse River Basin in the Columbia River Plateau provides a case study of the simple shift in perception of recharge as a consequence of a holistic integration of hydraulic testing, hydrogeochemical interpretations, and flow model parameters. The findings of 50 years of projects and new research scenarios in this heterogeneous and anisotropic aquifer of sedimentary and volcanic origin can be summarized as follows:

1.  The recent collection and interpretation of hydrologic tracers (noble gases and H, O, C, and S isotopes) across the basin and within the different formations of the alluvial and confined aquifer system revealed a snowmelt-mountain front recharge process. The snowmelt from the mountain reliefs in the eastern portion of the basin represents the principal input for groundwater recharge, which was validated through simulation of particle traces through this multiple aquifer system;

2.  Identification and recognition of paleochannels as coarser grained, highly permeable flowpaths originating along the mountain front shifted the perception from past ideas of very limited recharge or diffuse vertical recharge to the confined aquifer system. These architectural elements of fluvial origin represent the principal pathways for the snowmelt-mountain front recharge to the fractured basalt and interbedded sediment aquifer;

3.  The most recent MODFLOW model is based on a multidimensional physio-chemical dataset acquired that assisted in prescribing appropriate recharge parameters. Mod-

elled effective recharge of 25 and 105 mm/year appear to be appropriate for the basin plain and the mountain front, respectively. These values fit previous efforts on quantifying aquifer recharge based on chloride, storage, and mass-balance methods. This scenario contrasts a prior MODFLOW model that overestimated the recharge and underestimated hydraulic conductivity because of the perception of diffuse vertical recharge;

4.  Research on surface-groundwater interaction and aquifer recharge can be further advanced by using new EPM solutions that incorporate overland flow, unsaturated and groundwater flow given the pre-existence of a robust dataset at the study site.

Overall, the integration of results from hydrogeochemical investigation with groundwater flow and particle tracking modelling were needed to appropriately determine recharge rates and dynamics in a geologically complex basin with high heterogeneity and anisotropy. Such efforts can assist in more quickly understanding aquifer dynamics providing useful and quantitative information to resource managers.

**Author Contributions:** Conceptualization, G.M. and J.B.L.; methodology, G.M.; investigation, G.M. and J.B.L.; resources, G.M. and J.B.L.; data curation, G.M.; writing—original draft preparation, G.M.; writing—review and editing, J.B.L.; visualization, G.M.; supervision, J.B.L.; project administration, J.B.L. All authors have read and agreed to the published version of the manuscript.

**Funding:** This research received no external funding.

**Institutional Review Board Statement:** Not applicable.

**Informed Consent Statement:** Not applicable.

**Data Availability Statement:** Not applicable.

**Acknowledgments:** The research is a review inspired by a larger project that aims to ensure the sustainability of the regional groundwater resources of the South Fork Palouse River Basin. In this framework, the manuscript also benefitted from constructive discussions with John Bush (University of Idaho), and Jan Boll (Washington State University).

**Conflicts of Interest:** The authors declare no conflict of interest.

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
