# Peer review of "Pathways and Estimate of Aquifer Recharge in a Flood Basalt Terrain; A Review from the South Fork Palouse River Basin (Columbia River Plateau, USA)"

_sustainability, doi:10.3390/su141811349_

Round 1
Reviewer 1 Report
This manuscript titled: “Pathways and estimate of aquifer recharge in a flood basalt terrain; a review from the South Fork Palouse River Basin (Columbia River Plateau, USA)” is an insightful and innovative review which encourages hydrogeologists to combine physicochemical and physical signatures in characterising and constraining recharge. It does this by reviewing and gathering previous literature spanning the past 50+ years for the South Fork Palouse River Basin. It also espouses the method of going back to basics in recharge characterisation by encouraging the hydrogeological understanding that can be gleaned from topographical and geological conceptual models. The work is well and strongly referenced demonstrating the understanding and synthesis of recharge processes in the catchment of study.
General Concept Comments
The article is well written, logical, coherent and flows fluently in simple and well structured, and easy to understand English.
Review
The article is novel and innovative in that it succinctly presents the history of recharge efforts in the South Fork Palouse River Basin, and demonstrates and espouses the integrative use of physicochemical and physical hydrogeological methods for constraining recharge in the catchment. It follows the current trend of integrated studies using several lines of evidence in hydrogeological investigations. Of great importance in this work is that a range of possible recharge values and phenomenon can be occurring in the same catchment depending on the physical, topographical and geological controls on the hydrogeology. This idea and work can be replicated in other heterogenous catchments worldwide. The references in this review are current and relevant in relation to recharge studies and phenomenon.
Specific comments
The reviewer recommends the use of the word “integrated” rather than the word “intersection” as used at places by the authors. There are few restructuring of tense and lines that have been commented on in the manuscript.
Figures are ok, but for Figure 1, that needs the a,b and c inserted on the different plates to aid reading and comprehension.
Overall, content flows well and the arguments are logical and consistent. Conclusion follows the theme of the paper from the beginning to the end.

Author Response
Replies to Reviewer 1
General Concept Comments
The article is well written, logical, coherent and flows fluently in simple and well structured, and easy to understand English.
Review
The article is novel and innovative in that it succinctly presents the history of recharge efforts in the South Fork Palouse River Basin, and demonstrates and espouses the integrative use of physicochemical and physical hydrogeological methods for constraining recharge in the catchment. It follows the current trend of integrated studies using several lines of evidence in hydrogeological investigations. Of great importance in this work is that a range of possible recharge values and phenomenon can be occurring in the same catchment depending on the physical, topographical and geological controls on the hydrogeology. This idea and work can be replicated in other heterogenous catchments worldwide. The references in this review are current and relevant in relation to recharge studies and phenomenon.
Specific comments
The reviewer recommends the use of the word “integrated” rather than the word “intersection” as used at places by the authors. There are few restructuring of tense and lines that have been commented on in the manuscript.
Reply: We avoided to use the word “integrated” as suggested by the reviewer (Line 25)
Figures are ok, but for Figure 1, that needs the a,b and c inserted on the different plates to aid reading and comprehension.
Reply: Change made to Figure 1 as requested (Lines 98, 99)
Overall, content flows well and the arguments are logical and consistent. Conclusion follows the theme of the paper from the beginning to the end.
Reviewer 2 Report
Dear Authors,
you did a nice review for recharge on this basin. Please consider a few changes :
1- You may consider adding at line 54, references of Mattei et al (https://doi.org/10.1002/vzj2.20066 and doi:10.3390/w12020393) ;
2- Table 1 content should be reviewed according fig 1 units names (e.g. Deep Grande Ronde versus Laower Grande Ronde) and column "Epoch" need to be corrected (stage are replaced by tickness)
3- Are you sure that in equation (1), P is the pumping rate ?? I would be surprised...or if, please explain.
Author Response
Replies to Reviewer 2
Dear Authors,
You did a nice review for recharge on this basin. Please consider a few changes :
1- You may consider adding at line 54, references of Mattei et al (https://doi.org/10.1002/vzj2.20066 and doi:10.3390/w12020393)
Reply: Reference added as requested by the reviewer (Lines 56, 485-487)
2- Table 1 content should be reviewed according fig 1 units names (e.g. Deep Grande Ronde versus Lower Grande Ronde) and column "Epoch" need to be corrected (stage are replaced by thickness)
Reply: Changes now applied on unit names and geological time (Fig. 1; Table 1)
3- Are you sure that in equation (1), P is the pumping rate ?? I would be surprised...or if, please explain
Reply: P is the volume of water pumped. The equation has been changed under request of the reviewer (Line 254)
Reviewer 3 Report
Review for sustainability-1893075
Thank you for the opportunity to review this manuscript. The authors present an interesting review paper which addresses the challenges of assessing aquifer recharge in a complex hydrogeological settings of South Fork Palouse River Basin flood basalt terrains.
The main aim of this review paper was to illustrate the advantage of a holistic approach to the understanding of aquifer recharge in heterogeneous systems, such as the fractured basalt and interbedded sediments of the Columbia River Plateau.
To achieve the aim, the authors give a brief but comprehensive overview of the research methods that have been applied over the past 50 years to understand the recharge behavior of the complex hydrogeological system. By giving the overview, authors demonstrate their broad knowledge when opening the problem, while showing the fitting of their own comprehensive research results to the context. The recharge rates derived from several different approaches were compared, and the most successful approach was identified. The authors stress the need for a good conceptual understanding of hydrogeological system to create a reliable groundwater flow model. In addition, some bottlenecks are also discussed to be specified in the future.
In conclusion, the review article seems to be well written, compact and concise, and can be recommended for publication in your journal after minor revisions.
Below are some suggestions for improving the manuscript:
Lines 57-61: Maybe it's my lack in this field, but Ref. 26 seems to be relatively random there;
Line 66: Space missing;
Line 79: Ref. 30 features a case about San Piedro River in Arizone, while Ref. 38 seems to be a study on a loess plateau in China. It could be some kind of a misunderstanding here, but if not, I would suggest to carefully go through all the cited literature to ensure their relevance;
Lines 84-87: Couldn’t manage to spot anything about tracers and groundwater flow models in the cited references. Did I miss something?;
Lines 102-216: This whole section could use some additional rewriting and paraphrasing as it seems largely similar to https://doi.org/10.1007/s41742-021-00318-0;
Lines 135-136: Very similar looking figure in https://doi.org/10.1007/s41742-021-00318-0. For example, the "Quarries" symbol seems to serve no purpose in this manuscript. The figure could use some more alteration;
Line 297: Reference 78 missing in the list. Ref. no. 79 seems to be also missing;
Line 323: I am not totally convinced if the term “aerial infiltration” is a valid one here;
Line 356: letter “o” missing in “palaeochannels”.
Author Response
Replies to Reviewer 3
Thank you for the opportunity to review this manuscript. The authors present an interesting review paper which addresses the challenges of assessing aquifer recharge in a complex hydrogeological settings of South Fork Palouse River Basin flood basalt terrains.
The main aim of this review paper was to illustrate the advantage of a holistic approach to the understanding of aquifer recharge in heterogeneous systems, such as the fractured basalt and interbedded sediments of the Columbia River Plateau.
To achieve the aim, the authors give a brief but comprehensive overview of the research methods that have been applied over the past 50 years to understand the recharge behavior of the complex hydrogeological system. By giving the overview, authors demonstrate their broad knowledge when opening the problem, while showing the fitting of their own comprehensive research results to the context. The recharge rates derived from several different approaches were compared, and the most successful approach was identified. The authors stress the need for a good conceptual understanding of hydrogeological system to create a reliable groundwater flow model. In addition, some bottlenecks are also discussed to be specified in the future.
In conclusion, the review article seems to be well written, compact and concise, and can be recommended for publication in your journal after minor revisions.
Below are some suggestions for improving the manuscript:
Lines 57-61: Maybe it's my lack in this field, but Ref. 26 seems to be relatively random there
Reply: We changed the reference with one much more specific on aquifer recharge (Lines 506-509)
Line 66: Space missing
Reply: Space added as suggested by the reviewer (Line 66)
Line 79: Ref. 30 features a case about San Piedro River in Arizone, while Ref. 38 seems to be a study on a loess plateau in China. It could be some kind of a misunderstanding here, but if not, I would suggest to carefully go through all the cited literature to ensure their relevance;
Reply: Reference 30 changed with a new one specific for the Palouse area as suggested by the reviewer (Lines 516-519). The reference 38 is from the Palouse area.
Lines 84-87: Couldn’t manage to spot anything about tracers and groundwater flow models in the cited references. Did I miss something?
Reply: The literature on environmental tracers is complete for the specific site, the research was performed by profs. Keller and Langman. However, we added new literature on the soil properties in the Palouse (Lines 516-519)
Lines 102-216: This whole section could use some additional rewriting and paraphrasing as it seems largely similar to https://doi.org/10.1007/s41742-021-00318-0
Reply: The second author applied major re-writing on the geo-hydrological setting with respect to the test written by Medici et al (2021, IJER) before submitting the earlier version. However, we found that several sentences needed re-wording (Lines 104-105, 113, Line 116-123, Line 135, Line 152, Lines 161-208), a new figure has been added to this section (Figure 3c), and a comparison was added to the text between the Palouse and Tuscany regions (Lines 156-158)
Lines 135-136: Very similar looking figure in https://doi.org/10.1007/s41742-021-00318-0. For example, the "Quarries" symbol seems to serve no purpose in this manuscript. The figure could use some more alteration
Reply: The legend has been changed. Quarry is un-necessary detail as pointed-out by the reviewer. We also applied corrections to the name of the units to match the Table 1
Line 297: Reference 78 missing in the list. Ref. no. 79 seems to be also missing;
Reply: References added as suggested by the reviewer (Lines 617-622)
Line 323: I am not totally convinced if the term “aerial infiltration” is a valid one here
Reply: Change made, we now use the term “spatially uniform infiltration” (Line 323)
Line 356: letter “o” missing in “paleochannels”.
Reply: Correction made (Line 637)
Reviewer 4 Report
The manuscript is mostly a review of existing studies to determine recharge, with comments, and the proposal of a revised concentual model of mountain front recharge. The manuscript is mostly a summary, so the different approaches are not discussed nor the associated uncertainty is quantified. As a summary, the manuscript is useful to better understand the quantification on groundwater flow, but the reader have to resource to the well detailed references to understand what is said..
Some more detail on surface water recharge and snowmelt role along the year will be welcome, as well as information on the data for the chloride mass balance. Some of the mentioned recharge estimations derive from estudies at detailed scales that may be problematic when applied al a larger scale. Do hydrochemical studies agree with hat is said?
Give recommendation for future studies and reseach, especially common errors to be avoided and improved observations and monitoring.
There a few details:
146 says his instead of is
154-156 reconsider the sentence to avoid mixing water depth with snow thickness
220 236 343 says aerial; sholud it be areal?
231 236 says 12C and it should probablt 13C
239 give the age in the most common unit of years
255 Table 2 explain the meaning of potencial recharge and affective recharge
Author Response
Replies to Reviewer 4
The manuscript is mostly a review of existing studies to determine recharge, with comments, and the proposal of a revised conceptual model of mountain front recharge. The manuscript is mostly a summary, so the different approaches are not discussed nor the associated uncertainty is quantified. As a summary, the manuscript is useful to better understand the quantification on groundwater flow, but the reader have to resource to the well detailed references to understand what is said…
Some more detail on surface water recharge and snowmelt role along the year will be welcome, as well as information on the data for the chloride mass balance. Some of the mentioned recharge estimations derive from studies at detailed scales that may be problematic when applied al a larger scale. Do hydrochemical studies agree with hat is said?
Reply: Computed travel times by numerical modelling (basin scale) fit the groundwater ages constrained by using hydrochemistry on discrete points. This point was already explained in the previous version, we added more detail under request of the reviewer (Lines 359-370)
Give recommendation for future studies and research, especially common errors to be avoided and improved observations and monitoring.
Reply: Recommendation on new experimental efforts and more robust monitoring program now added on the new version of the manuscript (Lines 393-399)
There are a few details:
Line 146 says his instead of is
Reply: Correction made (Line 148)
Lines 154-156 reconsider the sentence to avoid mixing water depth with snow thickness
Reply: Clarification made (Line 161)
Lines 220, 236, 343 says aerial; sholud it be areal?
Reply: Clarifications made (Lines 226, 274, 308, 334)
Lines 231, 236 says 12C and it should probably 13C
Reply: Clarifications made (Lines 237, 242)
Line 239 give the age in the most common unit of years
Reply: We now also use years as suggested by the reviewer (Line 245)
Line 255, Table 2 explain the meaning of potential recharge and affective recharge
Reply: Explanation of the definition of potential and effective recharge added in the text as suggested by the reviewer (Lines 263, 276-277)